# Evaluating the Feasibility of Forward Osmosis in Diluting RO Concentrate Using Pretreatment Backwash Water

**DOI:** 10.3390/membranes10030035

**Published:** 2020-02-25

**Authors:** Susanthi Liyanaarachchi, Veeriah Jegatheesan, Li Shu, Ho Kyong Shon, Shobha Muthukumaran, Chun Qing Li

**Affiliations:** 1School of Engineering, RMIT University, Melbourne, VIC 3000, Australia; susanthiliya@gmail.com (S.L.); li.shu846@gmail.com (L.S.); chunqing.li@rmit.edu.au (C.Q.L.); 2School of Civil and Environmental Engineering, University of Technology Sydney, Broadway, NSW 2581, Australia; Hokyong.Shon-1@uts.edu.au; 3College of Engineering & Science, Victoria University, Melbourne, VIC 8001, Australia; Shobha.Muthukumaran@vu.edu.au

**Keywords:** biofouling, fertilizers, flat sheet, flux, forward osmosis (FO), hollow fibre, reverse osmosis (RO)

## Abstract

Forward osmosis (FO) is an excellent membrane process to dilute seawater (SW) reverse osmosis (RO) concentrate for either to increase the water recovery or for safe disposal. However, the low fluxes through FO membranes as well the biofouling/scaling of FO membranes are bottlenecks of this process requiring larger membrane area and membranes with anti-fouling properties. This study evaluates the performance of hollow fibre and flat sheet membranes with respect to flux and biofouling. Ferric hydroxide sludge was used as impaired water mimicking the backwash water of a filter that is generally employed as pretreatment in a SWRO plant and RO concentrate was used as draw solution for the studies. Synthetic salts are also used as draw solutions to compare the flux produced. The study found that cellulose triacetate (CTA) flat sheet FO membrane produced higher flux (3–6 L m^−2^ h^−1^) compared to that produced by polyamide (PA) hollow fibre FO membrane (less than 2.5 L m^−2^ h^−1^) under the same experimental conditions. Therefore, long-term studies conducted on the flat sheet FO membranes showed that fouling due to ferric hydroxide sludge did not allow the water flux to increase more than 3.15 L m^−2^ h^−1^.

## 1. Introduction

Diminishing freshwater resources pose a serious threat to various practices. For example, if the water required to make fertiliser solutions can be sourced from impaired water, it will significantly help to conserve freshwater sources for other activities in agricultural farms. Similarly, if the brine produced in a seawater reverse osmosis (SWRO) plant can be diluted using the pre-treatment filter backwash water, it can be reused as the feed to the RO or can be discharged safely to the receiving environment [1,2]. When a Forward Osmosis (FO) membrane separates diluted feed stream (impaired water) and concentrated draw stream (concentrated fertilizer or RO brine solutions), water will naturally pass through the FO membrane from the dilute stream to the concentrated stream to produce diluted solutions. This is due to the osmotic pressure difference created by those two streams. The larger the osmotic pressure difference, the higher the water flux through the membrane. At the same time, reverse salt flux, RSF (movement of salts from the concentrated stream to the dilute stream) would also occur, which can be minimised by selecting appropriate membranes. Generally, FO membranes will have active and support layers which will reduce the effective osmotic pressure difference between the two surfaces of the active layer and thus will reduce the water flux [3,4,5].

The FO membranes can be obtained either as flat sheets or hollow fibres. Flat sheet membranes available to date are showing lower water fluxes when the concentrations of draw solutions are low [1,6,7,8,9,10,11,12]. One of the best available cellulose triacetate (CTA) flat sheet membranes manufactured by Hydation Technology Innovations (HTI), USA, provides a maximum water flux of 9.6 L m^−2^ h^−1^ (LMH) when deionised (DI) water and 0.6 M NaCl salt solution were used as feed and draw solutions, respectively [13].

According to Li et al. [14], CTA FO membranes are made by adding dried CTA and cellulose acetate (CA) polymers to a premixed solvent of dioxane, acetone, lactic acid, and methanol at a certain ratio. The polymer/solvent solution will be kept at 30 °C and stirred till a homogeneous solution is obtained. The solution will then be stored in an oven at 30 °C for several hours to de-aerate and then will be cast onto a dry clean glass plate. The formed film will be immersed into a water bath within 3 s at 12 °C. After solidification, the membranes will be immersed in deionized water for 24 h before conducting any tests.

According to Lim et al. [15], a typical dry-jet wet spinning method can be applied for preparation of the hollow fibre membrane substrates. Dried polyether sulfone (PES) powder and polyethylene glycol (PEG400) at a fixed amount can be mixed with N-Methyl-2-pyrrolidone (NMP) at 60 °C for 12 h. Hydrophilic non-solvent (PEG) is added into the polymer solution to produce a sponge-like porous morphology for enhancing pore formation and interconnection. A degassed polymer solution will then be pumped into the double spinneret nozzle together with the bore fluid and the molded fibres will be immersed into the coagulation bath immediately. The solidified substrates will then be rolled up and stored in DI water for 24 h. The hollow fibre membranes will be immersed in the aqueous glycerol solution (50 wt%) for two days and dried in the atmosphere to minimise the collapse of their pore structures in open-air storage. Hollow fibre membrane modules can be made using the fibres [15].

This study evaluates the performance of a hollow fibre membrane over a flat sheet membrane with respect to the water flux. Based on the flux results, biofouling of flat sheet membranes was also evaluated.

## 2. Materials and Methods

### 2.1. Membranes

Flat sheet cellulose tri-acetate (CTA) membranes were purchased from HTI, USA. The support layer of the flat sheet membrane is made up of polyester mesh and average pore diameter is 0.74 nm [16]. Scanning Electron Microscopy (SEM) images of the flat sheet CTA membrane are given in Figure 1a–c. As Figure 1a shows [17], the membrane is on an embedded screen support. Figure 1b shows the support layer and the embedded mesh and Figure 1c is the active layer where water permeation happens.

Hollow fibre polyamide (PA) membranes used were fabricated at Samsung Cheil Industries Inc., South Korea and consist of a Sulphonated Polysulphone (SPSf) support layer. SEM images of the hollow fibre PA membrane are given in Figure 1d,e [18]. Figure 1d shows the thickness of the lumens with pores, and Figure 1e shows the lumens. CTA and PA membranes used were hydrophilic and hydrophobic, respectively.

### 2.2. Flux Studies with Hollow Fibre Membrane

Feed (either deionised (DI) water or Fe(OH)_3_ sludge representing impaired water) and draw solutions (NaCl, MgCl_2_, CaCl_2_, Na_2_SO_4_ and seawater reverse osmosis concentrate (ROC) representing concentrated solution) were passed through the polyamide (PA) hollow fibre FO membrane at different feed and draw Reynolds number (*Re*) ratios. Reynolds numbers were varied by changing the velocity of the feed and draw solutions. Sludge/DI water was circulated outside the hollow fibre membrane and the draw solution through the lumen side. Since the lumen side surface of the hollow fibre is the active layer, the experiments have been run at an active layer facing draw solution (AL-DS) mode. Even though the sludge particles may block the support side in this mode, cleaning the outer surface of the fouled hollow fibre membranes will be much easier compared to cleaning the inner lumen side of the membrane. The experimental set up is shown in Figure 2. Change in the weight of the draw solution was programmed to be stored in a data logger at one-minute time intervals. Experimental water flux ($J_{w,e}$) was determined by the following equation:(1)$$J_{w,e}=\frac{\mathrm{change}\;\mathrm{in}\;\mathrm{weight}\;\mathrm{of}\;\mathrm{the}\;\mathrm{draw}\;\mathrm{solution}\;\mathrm{in}\;\mathrm{time}\;\Delta t}{\mathrm{density}\;\mathrm{of}\;\mathrm{water}\;\times \mathrm{effective}\;\mathrm{membrane}\;\mathrm{area}\;\times \Delta t}$$

After 1 h of filtration, properties of the feed and draw solutions were measured. The membrane was cleaned using DI water prior to each experiment.

### 2.3. Fouling Studies with Flat Sheet Membranes

The draw and feed solutions were ROC and pre-treatment sludge (filter backwash water) from a seawater RO desalination plant, respectively. Thus, the solids content of pre-treatment sludge was varied from 2% to 8% of total solids (TS). This solids content represents the suspended solids that are removed from the filter during the backwash. This is because there are two types of pre-treatment sludge that can be generated in an RO desalination plant; the media filters used for the pre-treatment of seawater can be backwashed using either pre-treated seawater or ROC and can produce pre-treatment sludge with different total solids and ionic strength. However, the dewatered sludge available in the lab had 15% TS as received from Perth Seawater Desalination Plant (PSDP). Therefore, to obtain required TS contents of each pre-treatment sludge, 15% TS sludge was diluted using (i) pre-treated seawater (and the feed solution obtained was named as High EC with an EC of 45 mS/cm) and (ii) DI water (and the feed solution obtained was named as low EC with an EC of 1.5 mS/cm). Prepared feed and draw solutions were passed through the cellulose triacetate (CTA) flat sheet FO membrane at 0.04 m/s cross flow velocity in counter current flow configuration [19,20].

Pre-treatment sludge was circulated on the support side of the membrane (FO mode) and stirred at a constant rate during the experiment to eliminate settling of particles. Experimental set up was similar to that in Figure 2a. Photos of the hollow fibre membrane module as well as the experimental setup are shown in Figure 2b. Experiments were run at 20 ± 2 °C and triplicated at each operating condition. Again, change in the weight of the draw solution was programmed to be stored in a data logger at 5 min time intervals. Furthermore, three consecutive experimental setups (similar to Figure 2a) were run. Fouling behaviour on the FO membrane was examined after one day, four days, one week and five weeks. One experiment was run until the membrane was fully fouled (i.e., until no water flux observed). Water flux, conductivity, total organic carbon (TOC), and pH of each set up were monitored continuously using a data logger, electrical conductivity (EC) meter, TOC analyser, and pH meter, respectively.

All of the fouling experiments were run in semi-batch mode as the experiments were long-term runs, following the experimental procedure of Li et al. [21], i.e., when the draw solution had extracted 15% of water from the feed (150 mL), both draw and feed solutions were replaced with fresh 1L tank; TOC, pH, temperature, and EC of the replaced solutions were measured. Prior to each new experiment, three experimental setups were thoroughly cleaned to remove trace organic matter using the procedure given in the next section [22]. It is important to note that the short-time treatment with alkaline hypochlorite solution could improve the membrane performance slightly [23]. Accordingly, the hypochlorite degradation reaction of aromatic PA membrane involves a reversible and an irreversible chlorination. The reversible intermediate could be regenerated to initial amide by the treatment with alkaline before it rearranged to an irreversible product, thus partially improving the membrane performance.

### 2.4. Cleaning of FO Set-up to Remove Trace Organic Impurities Prior to Each Fouling Test

The following procedure was followed to clean the FO set-up:I.Recirculation of 0.5% sodium hypochlorite through the FO set-up for 2 h.II.Removal of trace organic matter by recirculating 5 mM ethylene di-amine tetra-acetic acid (EDTA) at pH 11 through the set-up for 30 min.III.Additional removal of trace organic matter by recirculating 2 mM sodium dodecyl sulphate (SDS) at pH 11 through the set-up for 30 min.IV.Sterilisation of the unit by recirculating 95% ethanol through the set-up for 1 h.V.Rinsing the unit with DI water (several times) to eliminate ethanol residue.

Once the filtration was complete, a known area of membrane was selected for analysis for cell count, SEM, TOC, adenosine triphosphate (ATP), and live/dead cell count analysis (Figure 2c). Membrane fouling is due to the deposition of chemical (organic/inorganic) species and biological growth on the membrane surface. Thus, cell count is an indicator of biological growth and hence the progress of biofouling of the membrane.

## 3. Results

The properties of synthetic draw solutions used in this study are given in Table 1. Out of the five synthetic draw solutions used (1.0 M NaCl, 1.0 M MgCl_2_, 1.0 M CaCl_2_, 1.0 M Na_2_SO_4_ and ROC), Na_2_SO_4_ has the largest density while MgCl_2_ has the largest viscosity; CaCl_2_ has the highest electrical conductivity. Properties of seawater (as feed) and reverse osmosis concentrate (ROC) as drawn are given in Table 2.

### 3.1. Effect of Re on the Water Flux

Figure 3 shows the water flux through hollow fibre FO membranes when DI water and salt solutions were used as feed and draw solutions, respectively. Draw *Re* was kept at 1000 and 2000 while feed *Re* was kept at 200, 450, and 1200. Thus, six experiments with different feed and draw *Re* were conducted. Water flux of up to 10 LMH was observed when a laminar condition (Re = 1000) prevailed in the flow of draw solution. The Na_2_SO_4_ draw solution gave the highest flux similar to that of MgCl_2_. This is interesting as 1.0 M MgCl_2_ has the highest osmotic pressure; however, when the *Re* is similar, 1.0 M Na_2_SO_4_ shows similar performance even though its osmotic pressure is lower. Furthermore, when the *Re* of draw solution flow was increased to become near turbulent (at *Re* = 2000), all three draw solutions showed better performance compared to MgCl_2_. The MgCl_2_ solution drew a maximum of 5.1 LMH when the feed *Re* was the highest (1200). Therefore, it is evident that, when selecting a draw solution, not only its osmotic pressure, but also its viscosity, density, and the crossflow velocity are affecting the performance in terms of water flux.

Despite the type of solution, the feed flow with a *Re* of 200 and the draw solution flow with a *Re* of 1000 gave the best performance in terms of water flux. Low *Re* of feed flow provide enough time for the water to pass through the membrane and high *Re* of the draw flow reduces the dilution of the draw solution as it takes away the water flux coming from the feed side of the membrane quicker. This allows the FO process to produce high flux under those conditions. Therefore, sludge dewatering experiments (detailed in the following section) were conducted at 200:1000 feed to draw an *Re* ratio.

Reverse salt flux of the membrane was determined by measuring the EC values of the feed solution. Since the feed solution was DI water, the change in EC was obviously due to the ions transported through the membrane from the draw solution. Figure 4 shows the RSF (or EC values of the feed solution) for each draw solution. NaCl shows the highest increase in RSF with time. Despite the *Re*, RSF is increasing with the filtration time. CaCl_2_ shows the lowest RSF (below 5 µS/cm). MgCl_2_ which showed lower water fluxes compared to the other salt solutions shows lower RSF—however, higher than that of CaCl_2_. In general, the RSF in hollow fibre membrane was found to be small which was also the case in the literature [24,25]. Addition of divalent ions into the NaCl draw solution reduced the RSF but did not affect the flux, and MgCl_2_ was found to be a better additive [18]. However, multi-valent cations can form complexes with organic and colloidal foulants and expedite fouling and cake-enhanced osmotic pressure [26,27,28,29].

### 3.2. Effect of Sludge Solids Content

Figure 5 shows the amount of dewatered sludge from PSDP required to produce pre-treatment sludge with different total solids concentration. Since seawater and DI water were used for dilution, the vertical gap between the two graphs should be the TS content of the seawater. Therefore, the vertical gap 3.374 (= 3.3964 − 0.0224) should be the TS% in the seawater used to prepare pre-treatment sludge. Since TDS of seawater is 30–35 g/L, 3.374 TS% appears acceptable. For FO dewatering applications through hollow fibre membranes, low EC sludge samples were chosen assuming lower EC (hence, higher EC difference between the feed and the draw solution) would give better performance with the membrane.

Since the same membrane was used for each experiment (after cleaning), before and after the two-hour sludge dewatering, baseline experiments were run with 0.5 M NaCl and DI water as draw and feed solutions, respectively. This was to check whether the membrane coupon had returned to the initial condition after cleaning. The results are shown in Figure 6. As Figure 6 illustrates, cleaning has taken the membrane back to the original condition. This means, since the sludge dewatering time was only two hours, the membrane was either not fouled or the fouling is nearly 100% reversible. However, to compare the water flux at each sludge solids content, averaged water fluxes were plotted in one graph as shown in Figure 7. As Figure 7 shows, the lowest sludge solids content led to the highest water flux, i.e., 3.6 LMH, whereas all the other sludge types showed a flux of 1.5–2.5 LMH. When sludge solids content increased, there was a slight drop in the water flux. With increase in solids content, the viscosity and the density of the sludge increase. Higher viscosity means lower *Re*, and higher density means higher *Re*; however, the combination of higher viscosity and higher density led to lower water permeation through the hollow fibre membranes. The effect of higher amount of solids content was dominant, and this would have increased the concentration polarisation (CP) effect as sludge passed through the porous side of the membrane leading to lower water flux.

In our previous study [30], we conducted flux experiments with 4.04% pre-treatment sludge as feed solution and ROC as draw solution. CTA flat sheet FO membrane (from HTI USA) was used in the study as well. The active layer facing feed solution (AL-DS) mode gave a water flux of around 3 L m^−2^ h^−1^, which is higher than the flux obtained under the same condition in this study using hollow fibre membranes (flux between 2 and 2.5 when the % TS sludge varied from 3.68 to 4.67 as shown in Figure 7). Therefore, we decided to use flat sheet membranes for further studies on fouling.

### 3.3. Fouling Studies with Flat Sheet Membranes

Fouling of FO membrane is an important factor which will hinder the effectiveness of the process if not understood well and controlled through appropriate mitigative steps. However, sometimes, fouling will be beneficial depending on the application of the FO process. For example, Valladares Linares et al. [31] studied the effect of fouling of FO membrane in the rejection of micro-pollutants. They demonstrated clearly that fouled FO membrane rejected the hydrophilic ionic compounds and hydrophobic neutral compounds highly. Rejection of hydrophilic neutral compounds reduced slightly. They concluded that using FO along with low-pressure RO system can provide a barrier to the passage of micro-pollutants present in the secondary wastewater effluent. Such findings on treating wastewaters with high fouling propensities have been treated by FO processes are reported in other studies as well [32,33]. In another study, both reversible and irreversible membrane fouling were absent during the FO experiments when the active layer of the FO membrane was facing the activated sludge solution [34]. One study noted that the flux was strongly dependent on the type of FO membrane used, implying the varying level of fouling on different membranes [35]. Osmotic membrane bioreactors [36,37] are used to improve the removal of organic and ammonium ions from wastewater. Fouling was controlled by osmotic backwash. Large amounts of nutrients, total and suspended solids in the centrate that emerge from a centrifuge that dewaters digested sludge in a wastewater treatment plant, generally returned back to the inlet of the treatment plant. This stream contributes significantly to the nitrogen and phosphorus load of the influent. Recovering the nutrients using an FO/RO system has been evaluated in a study [36], which indicated that CTA-FO membrane showed less fouling (cake layer compaction) compared to CTA-RO membrane due to the lack of applied pressure. Flux recovery after cleaning of the foulants was also high since it was easy to remove the foulants deposited on the membrane. Furthermore, recent advances in the modification of FO membranes through nano-modifiers have improved anti-fouling properties of the membranes significantly and an excellent review on this by Sun et al. [38] can be found in the literature. Discussions on nano-modifiers such as low dimensional carbon-based nanomaterials (carbon nanotubes, graphene oxides, and carbon nanofibers), other nanomaterials (halloysite nanotubes, boehmite, silica, zeolite, nano-CaCO_3_), and metal/metal-oxide nanoparticles (silver and TiO_2_) can be found in the above-mentioned review [38]. Our study evaluates the performance of unique feed solution on the fouling of FO membrane which can contribute valuable information to the desalination industry.

#### 3.3.1. Change in Water Flux

The water flux pattern with time is shown in Figure 8. Flux declined with filtration time due to two reasons (1) fouling and (2) dilution of the draw solution as draw solution was recirculated. However, flux increased when the draw and feed solutions were replaced with fresh solution. This increased flux was lower than the initial flux of the previous batch due to fouling of the membrane. Flux decline due to fouling is shown in red dashed lines in Figure 8c,d. After one week of filtration, the flux declined further in Figure 8c due to the thickened fouling layer deposited on the membrane. The layer may have contained microorganisms and salt deposits as both draw and feed solutions contain salt ions. However, as the EDX spectrums shown in Figure 9a(iii),b(iii), after one week of continuous filtration, the FO system showed only salt deposits. This fouling could easily be removed by providing regular flushes at high cross flow velocities as deposited layers were thin and loose.

Interestingly, as Figure 8d shows, after 300 h (about 2 weeks), the flux was increased once more; however, it was less than the initial water flux. This was repeated after about 650 h as well (around 4 weeks). After about 2 weeks, the loose salt deposit layer had formed and when its thickness increased, part of the loose layer could be readily removed by the increased cross-flow velocity (because when thickness reduces, velocity increases, as shown in Figure 9c–e).

In summary, around 50% reduction in water flux was observed due to fouling during five weeks of continuous filtration, without cleaning in between. This is mainly due to deposition of metals. After eight weeks of filtration, there was no water permeation (Figure 10a). Salt deposition on the FO membrane coupon filtered for eight weeks was higher compared to the FO membrane coupon filtered for five weeks (Figure 10b,c). With frequent cleaning with water, water flux can be brought back to initial value as fouling in FO membrane is reversible. A once a week cleaning cycle may be required for longer runs as live (and dead) cells were observed after one week of filtration on the membrane surface.

#### 3.3.2. Total Organic Carbon (TOC) Results

Draw and feed solutions were replaced with fresh draw and feed samples every 24 h. The used feed and draw solution TOC values were measured daily. Eight weeks of TOC results are reported in Figure 11a. During eight weeks of filtration, TOC of the feed and draw solutions fluctuated. Once the 1-day, 4-day, 1-week, 5-week and 8-week filtration runs were completed, a known area of membrane was selected (as shown in Figure 2c) and vortexed with DI water to extract the deposited fouling layer. The extracted liquid was used to analyse the TOC content per unit of membrane area (Figure 11b). TOC on the membrane surface has increased 10 mg/cm^2^ when the filtration time increased from one day to five weeks. In addition, microorganisms started to grow on the membrane surface after one week of continuous filtration. As shown in Figure 11b, live and dead cells were propagated over the membrane surface which then led to reduction of water flux. Therefore, membrane may be needing at least a weekly cleaning cycle to avoid this. In the eight weeks of the filtration trial, the TOC value was significantly low (only 10 mg/cm^2^), which is hard to explain why. All the experiments other than an eight-week filtration trail were triplicated. Therefore, another duplicate experiment for an 8-week trial would be required to confirm the TOC results.

### 3.4. Future Needs to Improve the Performance of FO Membranes

In order to improve the water flux in a FO process, the internal concentration polarisation (ICP) of solutes should be reduced [39]. ICP is due to the migration of solutes through the porous support layer to the interface between the active and support layers of the membrane. This will reduce the effective osmotic pressure difference between the draw and the feed solution and reduce the flux. One of the ways to reduce ICP is to reduce the structural parameter of the membrane, S defined by the following formula:S = KD = tτ/ε(2)
where K is the resistance to solute diffusion, D is the diffusion coefficient of the solute; t is the thickness, τ is tortuosity and ε is porosity of the support layer of the membrane. The smaller the S, the larger the water flux. Recently, Shan et al. [40] have developed FO membranes by linking graphene oxide (GO) with oxidized carbon nanotubes (OCNTs) through oxygen-containing groups to form water channels in the polyamide layer. Such membrane had a very low S (230 µm) and showed very high water flux. For such membrane, when 0.5 M NaCl and deionized water were used as draw solution and feed solution, respectively, the water fluxes were 114 and 84.6 LMH for AL-DS and active layer facing feed solution (AL-FS) modes, respectively. The salt fluxes were 5.17 and 3.4 gMH for AL-DS and AL-FS modes, respectively. Further improvement to such membrane was carried out by Kang et al. [41] by assembling graphene oxide (GO) and oxidized carbon nanotubes (OCNTs) with five bilayers on a polyethersulfone membrane to reduce the salt passage through the membrane. An excellent review on membranes similar to the above producing high water fluxes and low reverse salt fluxes is documented by Sun et al. [38].

## 4. Conclusions

Water fluxes produced by CTA flat sheet and PA hollow fibre FO membranes were compared to select appropriate membrane for further studies on fouling. The Reynolds Number (*Re)* of draw and feed solutions was varied to enhance the water flux through membrane. Lower *Re* of feed and draw solution flows produced better water flux compared to higher *Re* of feed and draw solution flows. When both membranes are used to derive water flux with pre-treatment sludge (or filter backwash water) as feed solution and ROC as draw solution, the PA hollow fibre membrane yielded an average water flux of 2.1 LMH. The process was operated under AL-DS mode, and the sludge solids content in the pretreatment sludge was 3.68%. In our previous study, under similar conditions, flat sheet CTA membranes showed 1.5 times higher water flux compared to PA hollow fibre membranes. Further studies on fouling using the CTA flat sheet membrane confirmed that water flux can decrease by 50% over a period of five weeks due to fouling, if the membrane is not cleaned in between. If the FO process is continued to run without further cleaning, the flux ceases after eight weeks from the beginning of the run. With frequent cleaning with water, water flux can be brought back to the initial value as fouling in the FO membrane is reversible. A once a week cleaning cycle may be required for longer runs and to prevent biofouling. The flux obtained through both PA hollow fibre and CTA flat sheet membranes is less than or equal to 3.15 LMH, which is not sufficient for the system to be economical. Improving the performance of FO membranes by reducing the structural parameter through the introduction of nanomodifiers to the membrane material is one of the ways to go. This will enhance the utilization of FO in places where freshwater resources are diminishing and therefore reusing concentrates and reducing their impacts on fresh water sources are of paramount importance.

## Figures and Tables

**Figure 1 membranes-10-00035-f001:**
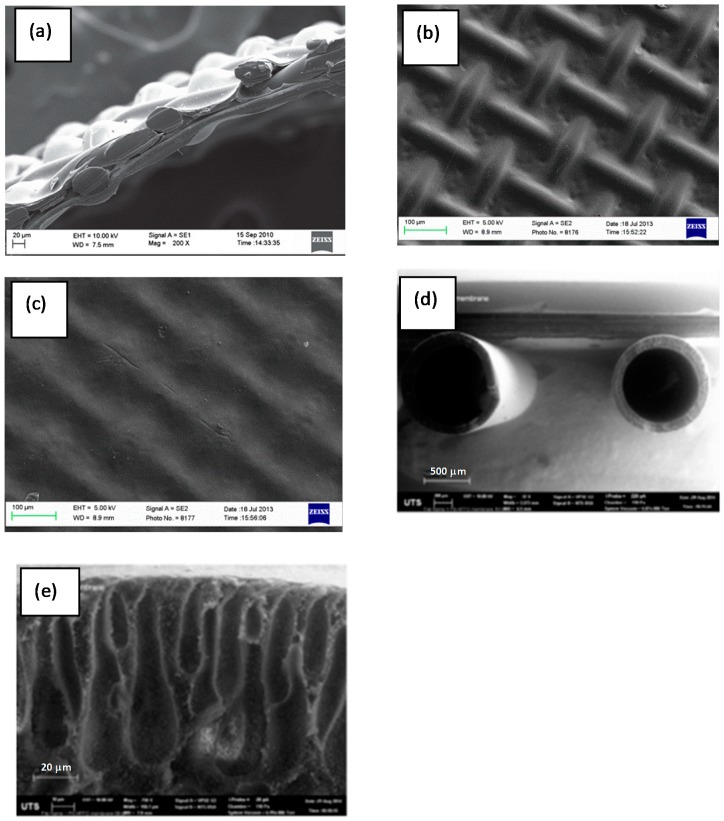
Images of hydrophilic Cellulose Triacetate (CTA) membrane on embedded polyester screen support: (**a**) cross section (reprinted from [17] with permission from Elsevier); (**b**) Support side; (**c**) active side. SEM images of hydrophobic Polyamide (PA) hollow fibre membrane (reprinted from [18] with permission from Elsevier) with inner surface of the hollow fibre as active layer, and the outer surface as support layer; (**d**) cross section showing the thickness of the lumens along with pores (**e**) enlarged cross section showing the lumen.

**Figure 2 membranes-10-00035-f002:**
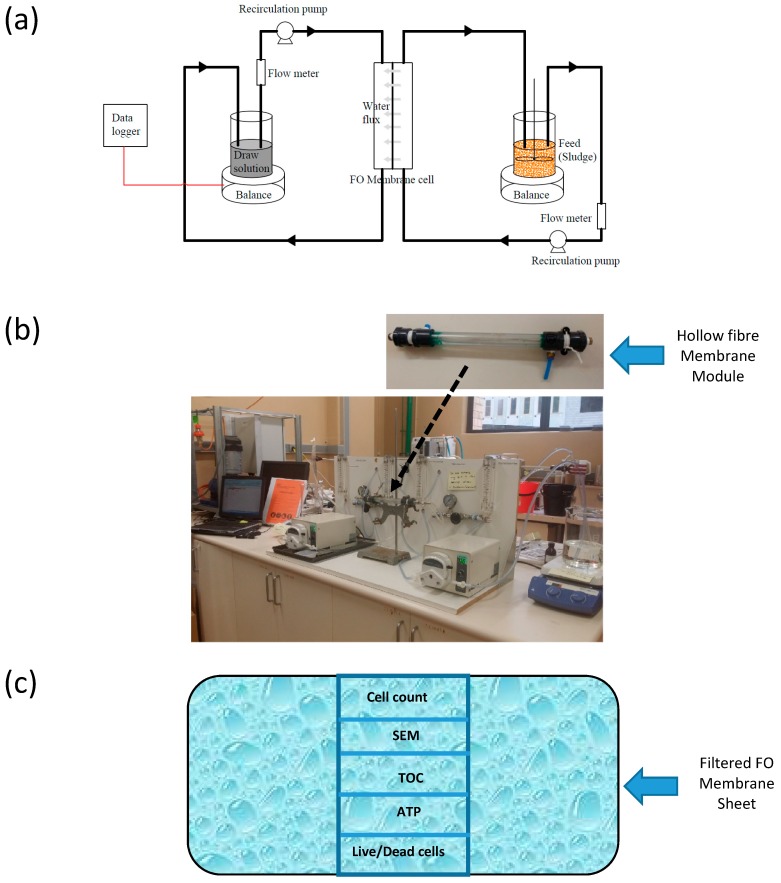
Hollow fibre membrane (**a**) schematic of the experimental set up with a general forward osmosis (FO) membrane cell that can accommodate either a flat sheet or a hollow fibre membrane module, (**b**) photo of the hollow fibre FO membrane system used in this study (Effective membrane area is 25.45 cm^2^), and (**c**) protocol for the analyses of fouled membrane.

**Figure 3 membranes-10-00035-f003:**
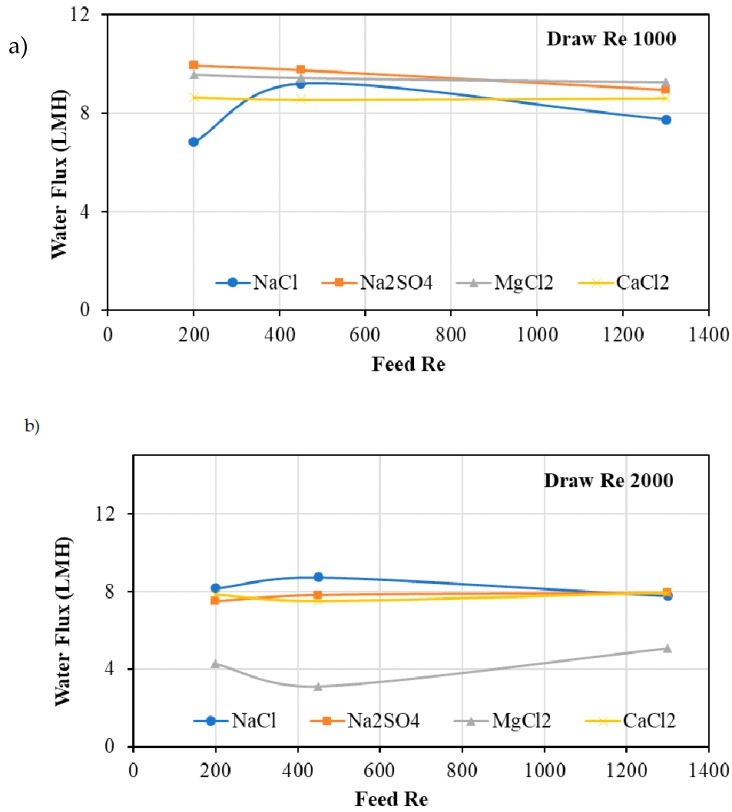
Flux through hollow fibre membranes when draw solution Re was (**a**) 1000 and (**b**) 2000. Note that the experiments were run in active layer facing draw solution (AL-DS) mode to compare the results with sludge dewatering experiments.

**Figure 4 membranes-10-00035-f004:**
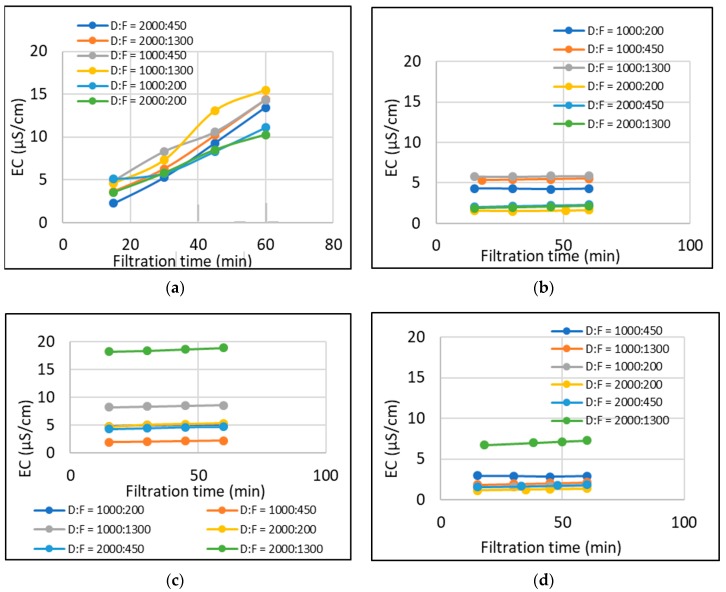
Measurements during filtration in hollow fibre FO membrane with different draw solutions (**a**) 1.0 M NaCl; (**b**) 1.0 M CaCl_2_; (**c**) 1.0 M MgCl_2_; (**d**) 1.0 M Na_2_SO_4_.

**Figure 5 membranes-10-00035-f005:**
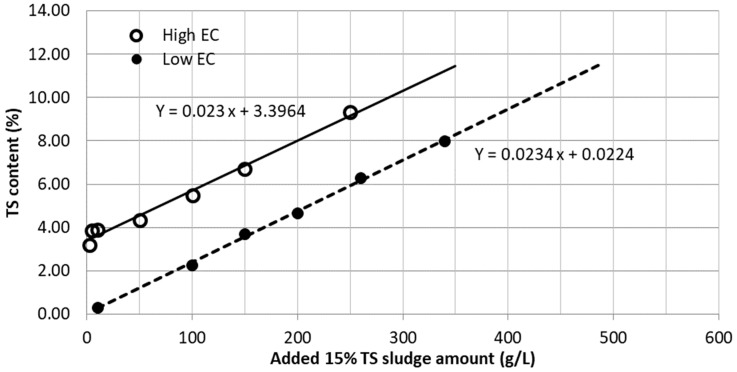
TS content of sludge with high and low EC (prepared in the laboratory starting with dewatered sludge having 15% TS obtained from PSDP).

**Figure 6 membranes-10-00035-f006:**
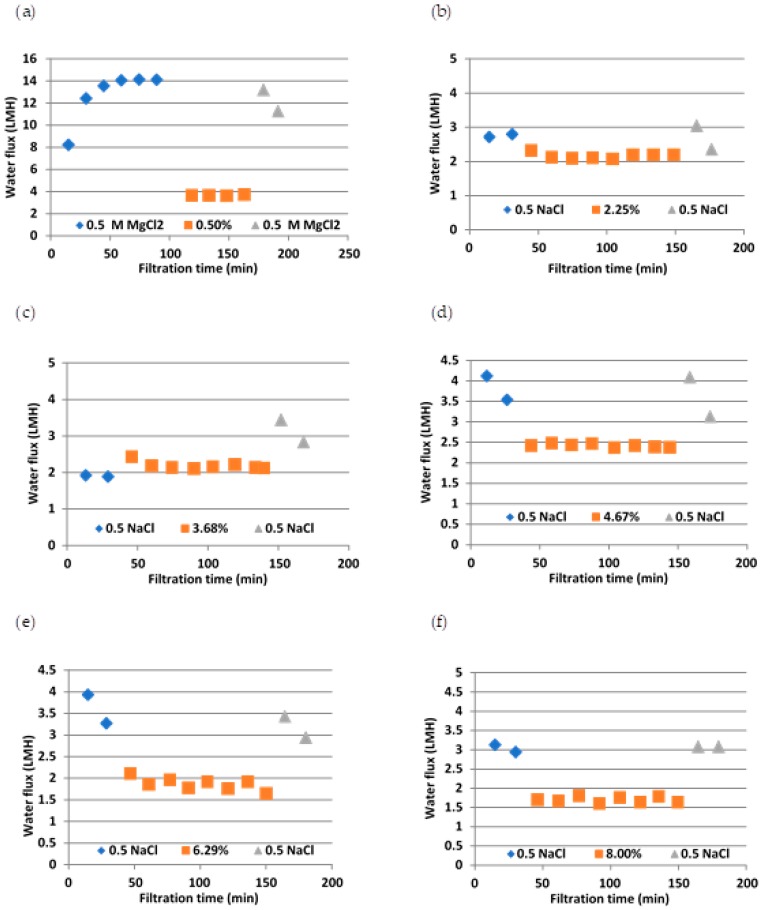
Flux at different sludge contents. Experiments were conducted at first with either MgCl_2_ (**a**) or NaCl (**b**–**f**) as draw solutions and DI as feed solution (shown in ♦) and then different concentrations of sludge were used as feed solution and reverse osmosis concentrate (ROC) as draw solution (shown in ■), and, finally, the membrane was cleaned and experiments were reverted back to original MgCl_2_ or NaCl as draw solutions and DI as feed solution (shown in ▲).

**Figure 7 membranes-10-00035-f007:**
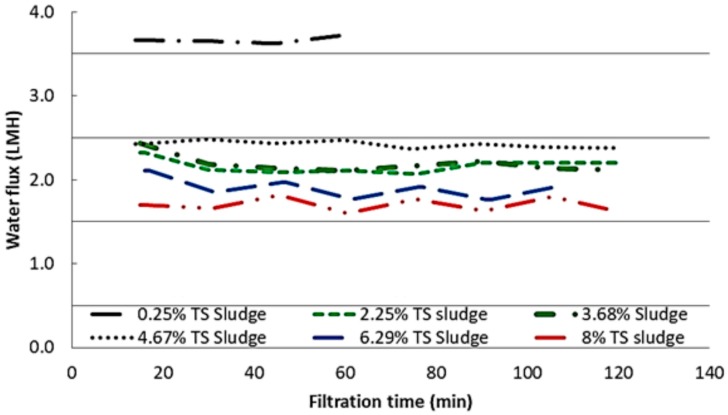
Comparison of water flux at each sludge solids concentrations.

**Figure 8 membranes-10-00035-f008:**
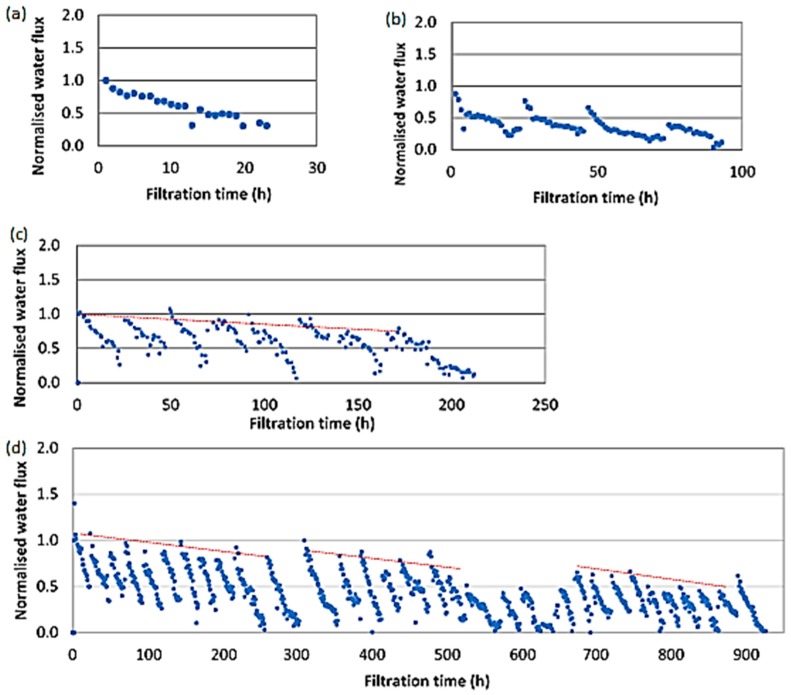
Flux through FO membrane during long-term filtration (**a**) 1 day; (**b**) 4 days; (**c**) 1 week; and (**d**) 5 weeks.

**Figure 9 membranes-10-00035-f009:**
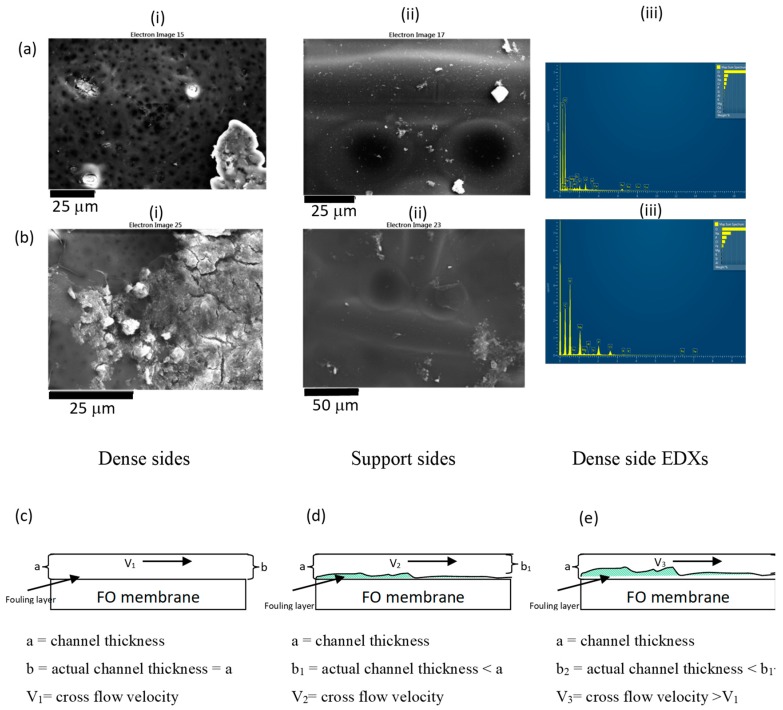
SEM images and EDX spectra of the membrane surface after (**a**) one week, (**b**) five weeks of filtration; schematic diagram of the FO membrane surface with filtration time: (**c**) initially, (**d**) during the first two weeks of operation, and (**e**) after about two weeks of operation. Fouling layer thickness increased over time so does the cross-flow velocity.

**Figure 10 membranes-10-00035-f010:**
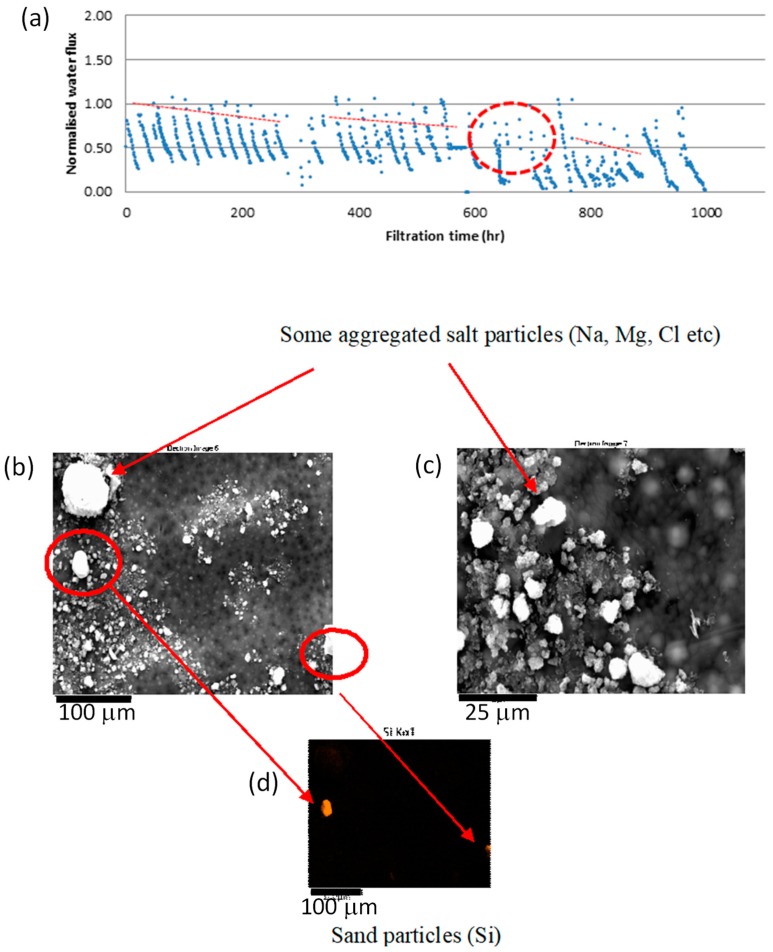
(**a**) normalised water flux with respect to filtration time; SEM images of the fouled membrane; (**b**) feed side; (**c**) draw side; (**d**) elemental analysis corresponding to (**a**) obtained through EDX.

**Figure 11 membranes-10-00035-f011:**
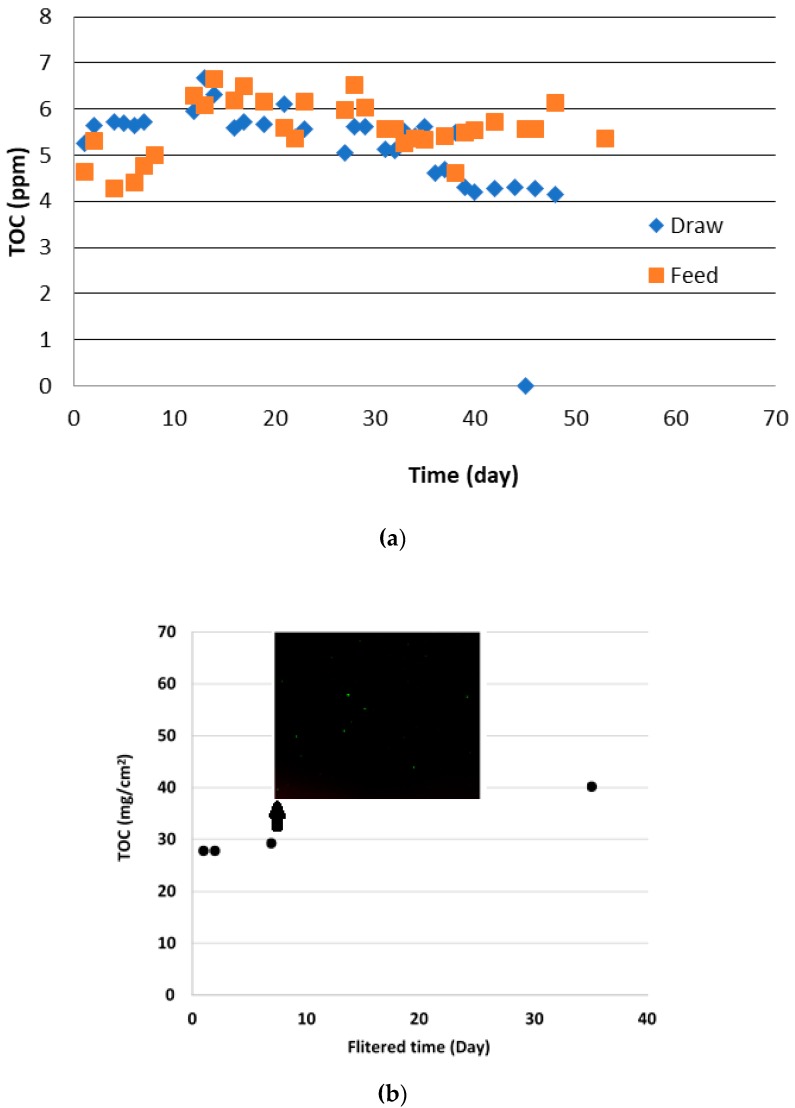
(**a**) Daily TOC results of the feed and draw solution; (**b**) TOC of the filtered membrane and live and dead cells on the membrane (Green—live cells and Red—dead cells).

**Table 1 membranes-10-00035-t001:** Properties of synthetic draw solutions used in this study.

Draw Solution (1 M)	Density, ρ (kg/m^3^)	Viscosity, µ (Pa∙s)	Conductivity *, EC (mS/cm)	Osmotic Pressure at 25 °C (bar)
NaCl	1037.00	0.001080	81.1	46.4
Na_2_SO_4_	1557.00	0.001120	81.9	52.0
MgCl_2_	1072.40	0.001490	96.7	79.9
CaCl_2_	1085.20	0.001330	108.6	80.0
ROC	1023.98	0.001004	72.3	33.0

Note: Density, viscosity and osmotic pressure were obtained from the OLI ^®^ stream analyser, and * conductivity from experimental values.

**Table 2 membranes-10-00035-t002:** Properties of feed and draw solution used in this study.

Property	Seawater	Sand Filtered Seawater	Draw Solution—ROC	Feed Solution—PSDP Fe(OH)_3_ Sludge
pH	8.42	7.68	7.77	8.69
Turbidity (NTU)	29.1	0.45	-	-
EC (mS/m)	4450	4470	7300	5150
TOC (mg/L)	1.71	0.73	3.10	17.06
Alkalinity—mg/L as CaCO_3_	110	45	68	102
Hardness (EDTA)-mg/L as CaCO_3_	4600	6200	9550	4500
Solids content (% TS)	-	-	-	4.04
Specific gravity	-	-	-	1.01

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
