# Peer review of "Evaluating the Feasibility of Forward Osmosis in Diluting RO Concentrate Using Pretreatment Backwash Water"

_membranes, 2020, doi:10.3390/membranes10030035_

Round 1

Reviewer 1 Report

In this manuscript, you evaluated the performance of a hollow fibre membrane over a flat sheet membrane with respect to the water flux. Based on the flux results, biofouling of flat sheet membranes was also evaluated. Although plenty of data were presented, there were still some shortcomings in the paper.

My comments are as follows:

  1. In order to make the article more perfect, it is hoped to add the preparation method of the FO membrane, and in the first sentence of the abstract, the word “Osmosis” does not need to be capitalized.
  2. In the page 2, “One of the best available CTA flat sheet membranes manufactured by HTI innovations USA provides a maximum water flux of 9.6 LMH when DI water and 0.6 M NaCl salt solution were used as feed and draw solutions, respectively.”, the abbreviation for the first use of the letter CTA did not explain its meaning.
  3. In the page 3, “Removal of trace organic matter by recirculating 5 mM ethylene di-amine tetra-acetic acid (EDTA) at pH 11 through the set-up for 30 min.” and “Additional removal of trace organic matter by recirculating 2 mM sodium dodecyl sulphate (SDS) at pH 11 through the set-up for 30 min.”, the millimeter units in these two sentences are not standard.
  4. Please indicate the reason for the cell count analysis.
  5. Please explain the abbreviations of ATP and TOC.
  6. The note in the name of Figure 2 will be more standard under the figure, such as Table 1.
  7. The position of the abscissa text in Figure 3 (a) is on the picture, which is not standardized.
  8. The name of Figure 8 is not standardized and the label of (a) in Figure 9 is missing.
  9. In the page 7,display errors in lines 17 to 18.
  10. In 3.3 Fouling studies with flat sheet membranes of section, please strengthen the mechanism analysis and comment of membrane fouling, some references could be referred, such as, 10.1016/j.jcis.2019.04.014; 10.1016/j.apsusc.2018.09.255ï¼›10.1039/C7RA12835E
  11. In the section "3.3. Fouling studies with flat sheet membranes", the cited literature does not have a common central theme and the transition is stiff.

Author Response

First of all we thank the reviewer for providing valuable comments which have improved the manuscript significantly. The corrections and inclusions in the revised manuscript are highlighted in red color text.

1. In order to make the article more perfect, it is hoped to add the preparation method of the FO membrane, and in the first sentence of the abstract, the word “Osmosis” does not need to be capitalized.

  • We did not prepare FO membranes. However, we understand the need to include the preparation method of FO membrane as the reviewer pointed out. Therefore, a summary on how the flat sheet and hollow fibre membranes can be made is provided with reference in the introduction. The following write-up is added in the introduction:

According to Li et al. [14], CTA FO membranes are made by adding dried CTA and cellulose acetate (CA) polymers to a premixed solvent of dioxane, acetone, lactic acid, and methanol at a certain ratio. The polymer/solvent solution will be kept at 30ËšC and stirred till a homogeneous solution is obtained. The solution will then be stored in an oven at 30ËšC for several hours to de-aerate and then will be cast onto a dry clean glass plate. The formed film will be immersed into a water bath within 3 s at 12ËšC.  After solidification, the membranes will be immersed in deionized water for 24 h before conducting any tests.

According to Lim et al. [15], a typical dry-jet wet spinning method can be applied for preparation of the hollow fibre membrane substrates. Dried polyether sulfone (PES) powder and polyethylene glycol (PEG400) at a fixed amount can be mixed with N-Methyl-2-pyrrolidone (NMP) at 60 °C for 12 h. Hydrophilic non-solvent (PEG) is added into the polymer solution to produce a sponge-like porous morphology for enhancing pore formation and interconnection. Degassed polymer solution will then be pumped into the double spinneret nozzle together with the bore fluid and the moulded fibres will be immersed into the coagulation bath immediately. The solidified substrates will then be rolled up and stored in DI water for 24 h. The hollow fibre membranes will be immersed in the aqueous glycerol solution (50 wt%) for two days and dried in the atmosphere to minimize the collapse of their pore structures in open-air storage. Hollow fibre membrane modules can be made using the fibres [15].

  • The word “Osmosis” is now not capitalized in the revision.

2. In the page 2, “One of the best available CTA flat sheet membranes manufactured by HTI innovations USA provides a maximum water flux of 9.6 LMH when DI water and 0.6 M NaCl salt solution were used as feed and draw solutions, respectively.”, the abbreviation for the first use of the letter CTA did not explain its meaning.

The abbreviation is expanded in the revision: Cellulose triacetate (CTA)

3. In the page 3, “Removal of trace organic matter by recirculating 5 mM ethylene di-amine tetra-acetic acid (EDTA) at pH 11 through the set-up for 30 min.” and “Additional removal of trace organic matter by recirculating 2 mM sodium dodecyl sulphate (SDS) at pH 11 through the set-up for 30 min.”, the millimeter units in these two sentences are not standard.

Kindly note that mM denotes milli-molar solution.

4. Please indicate the reason for the cell count analysis.

Membrane fouling is due to the deposition of chemical (organic/inorganic) species and biological growth on the membrane surface. Thus, cell count is an indicator of biological growth and hence the progress of biofouling of the membrane. This is included in the revised manuscript under section 2.3.

5. Please explain the abbreviations of ATP and TOC.

The abbreviations are expanded in the revision: adenosine triphosphate (ATP) and total organic carbon (TOC).

6. The note in the name of Figure 2 will be more standard under the figure, such as Table 1.

Discussions on experimental results are only starting from Table 2 and therefore we consider the note is in the appropriate location.

7. The position of the abscissa text in Figure 3 (a) is on the picture, which is not standardized.

It is corrected in the revised manuscript.

8. The name of Figure 8 is not standardized and the label of (a) in Figure 9 is missing.

Figure 8: We have labelled all the figures and have added the following note: Note: a(i) and b(i) dense sides of the membrane; a(ii) and b(ii) support sides of the membrane; a(iii) and b(iii) dense sides of the membrane.

Figure 9: Label (a) is added.

9. In the page 7, display errors in lines 17 to 18.

It should read as Figure 6 and it is corrected.

10. In 3.3 Fouling studies with flat sheet membranes of section, please strengthen the mechanism analysis and comment of membrane fouling, some references could be referred, such as, 10.1016/j.jcis.2019.04.014; 10.1016/j.apsusc.2018.09.255ï¼›10.1039/C7RA12835E

Thank you for this valuable suggestion which has allowed to improve the contribution of our manuscript with respect to future direction on membrane synthesis. The following has been included in a new section (3.4) of the revised manuscript that discusses about how to improve the flux by reducing the internal concentration polarisation which is essential to design an economical FO system:

3.4. Future needs to improve the performance of FO membranes

In order to improve the water flux in a FO process, the internal concentration polarisation (ICP) of solutes should be reduced [39]. ICP is due to the migration of solutes through the porous support layer to the interface between the active and support layers of the membrane. This will reduce the effective osmotic pressure difference between the draw and the feed solution and reduce the flux. One of the ways to reduce ICP is to reduce the structural parameter of the membrane, S defined by the following formula:

S = KD = tτ/ε                       (2)

Where, K is the resistance to solute diffusion, D is the diffusion coefficient of the solute; t is the thickness, τ is tortuosity and ε is porosity of the support layer of the membrane. The smaller the S, the larger the water flux. Recently, Shan et al. [40] has developed FO membranes by linking graphene oxide (GO) with oxidized carbon nanotubes (OCNTs) through oxygen-containing groups to form water channels in the polyamide layer. Such membrane had a very low S (230 μm) and showed very high water flux. For such membrane, when 0.5 M NaCl and deionized water were used as draw solution and feed solution respectively, the water fluxes were 114 and 84.6 LMH for AL-DS and active layer facing feed solution (AL-FS) modes respectively. The salt fluxes were 5.17 and 3.4 gMH for AL-DS and AL-FS modes respectively. Further improvement to such membrane was carried out by Kang et al.[41] by assembling graphene oxide (GO) and oxidized carbon nanotubes (OCNTs) with 5 bilayers on a polyethersulfone membrane to reduce the salt passage through the membrane. An excellent review on membranes similar to the above producing high water fluxes and low reverse salt fluxes are documented by Sun et al. [38].

11. In the section "3.3. Fouling studies with flat sheet membranes", the cited literature does not have a common central theme and the transition is stiff.

We agree to this comment and therefore went through the articles suggested by the reviewer and found the review article 10.1039/C7RA12835E to be of very useful to our manuscript. We have added the following in section 3,3 without much elaborating as ample information can be obtained from that review.

Further, recent advances in the modification of FO membranes through nano-modifiers have improved anti-fouling properties of the membranes significantly and an excellent review on this by Sun et al. [38] can be found in the literature. Discussions on nano-modifiers such as low dimensional carbon-based nanomaterials (carbon nanotubes, graphene oxides, carbon nanofibers), other nanomaterials (halloysite nanotubes, boehmite, silica, zeolite, nano-CaCO3) and metal/metal-oxide nanoparticles (silver and TiO2) can be found in the above-mentioned review [38].

Reviewer 2 Report

Should be “The larger the…, the higher the…” Section 2.2. Why would the DI water contain TS of 1.5 mS/cm? What does the author mean by “15% TS sludge was diluted using (i) pre-treated seawater (named as High EC with an EC of 45 mS/cm) …? Why did the authors selected the AL-DS mode (considering that the sludge particles may block the support side in this mode) ? Section 2.3. What was effect of the 0.5% sodium hypochlorite on the polyamide selective layer? Can the authors provide Electronic microscopic details and fabrication details of the hollow fiber FO membrane used? Table 1. Can the authors add a column of the osmotic pressure of the respective draw solutions? Section 3.2. Line 17. An error in reference. Figure 6. There appears to be a repetition of text in legend. Section 3.3. I am a bit confused why the authors had to select a different membrane (i.e., the flat sheet membrane) for the fouling test? Figure 10. TOC results appear to be faulty. As there is no data points between 10 -30 days.

Author Response

First of all, we thank the reviewer for providing valuable comments to improve our manuscript. All the inclusions and revisions in the revised manuscript are highlighted in red color font.

1. Should be “The larger the…, the higher the…”

It is corrected in the revision.

2. Section 2.2. Why would the DI water contain TS of 1.5 mS/cm?

Actually, sludge with 15% TS was diluted using DI water to bring the final EC to be 1.5 mS/cm. This has been now revised.

3. What does the author mean by “15% TS sludge was diluted using (i) pre-treated seawater (named as High EC with an EC of 45 mS/cm) …?

Again, sludge with 15% TS was diluted to obtain feed solution that can represent filter backwash water in a RO plant when pre-treated seawater was used to backwash. This is also now explained in the revision.

To answer the above two comments, the following revision was made to the relevant section:

The draw and feed solutions were ROC and pre-treatment sludge (filter backwash water) from a seawater RO desalination plant, respectively. Thus, the solids content of pre-treatment sludge was varied from 2 to 8 % of the total solids (TS). This solids content represents the suspended solids that are removed from the filter during the backwash. This is because there are two types of pre-treatment sludge can be generated in a RO desalination plant; the media filters used for the pre-treatment of seawater can be backwashed using either pre-treated seawater or ROC and can produce pre-treatment sludge with different total solids and ionic strength. However, the dewatered sludge available in the lab had 15% TS as received from Perth Seawater Desalination Plant (PSDP). Therefore, to obtain required TS contents of each pre-treatment sludge, 15% TS sludge was diluted using (i) pre-treated seawater (and the feed solution obtained was named as High EC with an EC of 45 mS/cm) and (ii) DI water (and the feed solution obtained was named as low EC with an EC of 1.5 mS/cm).

4. Why did the authors selected the AL-DS mode (considering that the sludge particles may block the support side in this mode)?

As we mentioned in our manuscript that “Since the lumen side surface of the hollow fibre is the active layer, the experiments have been run at active layer facing draw solution (AL-DS)”. Even though the sludge particles may block the support side in this mode, cleaning the outer surface of the fouled hollow fibre membranes will be much easier compared to cleaning the inner lumen side of the membrane. We have added this sentence in the revision.

5. Section 2.3. What was effect of the 0.5% sodium hypochlorite on the polyamide selective layer?

This is a very useful question and from the literature we found the following. This has been included in the revised manuscript:

It is important to note that the short-time treatment with alkaline hypochlorite solution could improve the membrane performance slightly [23]. Accordingly, the hypochlorite degradation reaction of aromatic PA membrane involves a reversible and an irreversible chlorination. The reversible intermediate could be regenerated to initial amide by the treatment with alkaline before it rearranged to irreversible product, thus partially improved the membrane performance.

6. Can the authors provide Electronic microscopic details and fabrication details of the hollow fiber FO membrane used?

  • We have not carried out SEM analysis but were able to include the SEM images of the membranes we used from other references with permission. Please see Figure 1 in the revised manuscript.
  • Fabrication of hollow fibre membrane: This question was asked by another reviewer as well. We did not prepare FO membranes. However, we understand the need to include the preparation method of FO membrane as the reviewer pointed out. Therefore, a summary on how the flat sheet and hollow fibre membranes can be made is provided with reference in the introduction. The following write-up is added in the introduction:

According to Li et al. [14], CTA FO membranes are made by adding dried CTA and cellulose acetate (CA) polymers to a premixed solvent of dioxane, acetone, lactic acid, and methanol at a certain ratio. The polymer/solvent solution will be kept at 30ËšC and stirred till a homogeneous solution is obtained. The solution will then be stored in an oven at 30ËšC for several hours to de-aerate and then will be cast onto a dry clean glass plate. The formed film will be immersed into a water bath within 3 s at 12ËšC.  After solidification, the membranes will be immersed in deionized water for 24 h before conducting any tests.

According to Lim et al. [15], a typical dry-jet wet spinning method can be applied for preparation of the hollow fibre membrane substrates. Dried polyether sulfone (PES) powder and polyethylene glycol (PEG400) at a fixed amount can be mixed with N-Methyl-2-pyrrolidone (NMP) at 60 °C for 12 h. Hydrophilic non-solvent (PEG) is added into the polymer solution to produce a sponge-like porous morphology for enhancing pore formation and interconnection. Degassed polymer solution will then be pumped into the double spinneret nozzle together with the bore fluid and the molded fibres will be immersed into the coagulation bath immediately. The solidified substrates will then be rolled up and stored in DI water for 24 h. The hollow fibre membranes will be immersed in the aqueous glycerol solution (50 wt%) for two days and dried in the atmosphere to minimize the collapse of their pore structures in open-air storage. Hollow fibre membrane modules can be made using the fibres [15].

7. Table 1. Can the authors add a column of the osmotic pressure of the respective draw solutions?

It is added in the revised manuscript.

8. Section 3.2. Line 17. An error in reference.

It is corrected and will read as Figure 6 in the revised manuscript.

9. Figure 6. There appears to be a repetition of text in legend.

This is fixed now.

10.Section 3.3. I am a bit confused why the authors had to select a different membrane (i.e., the flat sheet membrane) for the fouling test?

We selected flat sheet membrane as the flux obtained from hollow-fibre membrane was lower that the flux obtained from the flat sheet membrane. We have mentioned this just above the materials and methods section:

Based on the flux results, biofouling of flat sheet membranes was also evaluated.

We have also include the following section just before the section 3.3 on Fouling studies with flat sheet membranes:

In our previous study [31], we conducted flux experiments with 4.04% pre-treatment sludge as feed solution and ROC as draw solution. CTA flat sheet FO membrane (from HTI USA) was used in the study as well. The AL-FS mode gave a water flux of around 6 L m-2 h-1 which is higher than the flux obtained under the same condition in this study using hollow fibre membranes (flux between 2 and 2.5 when the % TS sludge varied from 3.68 to 4.67 as shown in Figure 7). Therefore, we decided to use flat sheet membranes for further studies on fouling.

11. Figure 10. TOC results appear to be faulty. As there is no data points between 10 -30 days. 

We ran fouling study for 1 day, 4 days, 1 week and 5 weeks and took the membrane swabs after those periods. Therefore we were unable to collect TOC samples in between weeks 1 and 5.  

Reviewer 3 Report

The review comments for membranes-727270

This work investigated the membrane fouling and water flux mitigation in the forward osmosis process when treating the RO concentrated brine. Synthetic salts were prepared as the draw solution to find the interrelation between water flux declining and organic fouling/inorganic deposition. The reversible and irreversible fouling/scaling was discussed to evaluate possible methods to refresh the FO membrane and revitalize the membrane’s original performance. As an original work in the membrane processes field, the study in work provides a reliable reference to the FO application. The work is publishable after a minor revision. I do not have scientific comments on the work, and only several writing problems were listed below for the author’s consideration.

  1. Please revise the abstract to highlight the innovations in the work.
  2. Reverse osmosis (RO) is suggested to substitute one keyword.
  3. The full names of abbreviations in the introduction should be supplemented, e.g., CTA, HTI, LMH.
  4. Page 7 line 17-18, error on reference citation.
  5. The discussion in conclusion and section 3.3.1 is repeated, please check it.

Author Response

First of all, we thank the reviewer for providing very useful comments to improve the manuscript. We have highlighted the revisions in red font in the revised manuscript.

1. Please revise the abstract to highlight the innovations in the work.

This is very important suggestion and we have include the following to highlight our innovations:

Forward osmosis (FO) is an excellent membrane process to dilute seawater (SW) reverse osmosis (RO) concentrate for either to increase the water recovery or for safe disposal. However, the low fluxes through FO membranes as well the biofouling/scaling of FO membranes are bottlenecks of this process requiring larger membrane area and membranes with anti-fouling properties. This study evaluates the performance of hollow fibre and flat sheet membranes with respect to flux and biofouling. Ferric hydroxide sludge was used as impaired water mimicking the backwash water of a filter that is generally employed as pretreatment in a SWRO plant and RO concentrate was used as draw solution for the studies. Synthetic salts are also used as draw solutions to compare the flux produced. The study found that cellulose triacetate (CTA) flat sheet FO membrane produced higher flux (3-6 Lm-2h-1) compared to that produced by polyamide (PA) hollow fibre FO membrane (less than 2.5 Lm-2h-1) under same experimental conditions. Therefore, long term studies conducted on the flat sheet FO membranes showed that fouling due to ferric hydroxide sludge did not allow the water flux to increase more than 3.15 L m-2 h-1.

2. Reverse osmosis (RO) is suggested to substitute one keyword.

RO is included as one keyword.

3. The full names of abbreviations in the introduction should be supplemented, e.g., CTA, HTI, LMH.

They all have been expanded when they appear for the first time in the text.

4. Page 7 line 17-18, error on reference citation.

This is corrected and should read as Figure 6.

It is corrected.

5. The discussion in conclusion and section 3.3.1 is repeated, please check it.

Thank you for this comment. We have checked and revised the conclusions:

Water fluxes produced by CTA flat sheet and PA hollow fibre FO membranes were compared to select appropriate membrane for further studies on fouling. The Reynolds Number (Re) of draw and feed solutions was varied to enhance the water flux through membrane. Lower Re of feed and draw solution flows produced better water flux compared to higher Re of feed and draw solution flows . When both membranes are used derive water flux with pre-treatment sludge (or filter backwash water) as feed solution and ROC as draw solution, the PA hollow fibre membrane yielded an average water flux of 2.1 LMH. The process was operated under AL-DS mode and the sludge solids content in the pretreatment sludge was 3.68%. In our previous study, under similar conditions, flat sheet CTA membranes showed 1.5 times higher water flux compared to PA hollow fibre membranes. Further studies on fouling using the CTA flat sheet membrane confirmed that water flux can decrease by 50% over a period of 5 weeks due to fouling, if the membrane is not cleaned in between. If the FO process is continued run without further cleaning, the flux ceases after 8 weeks from the beginning of the run. With frequent cleaning with water, water flux can be brought back to initial value as fouling in FO membrane is reversible. Once a week cleaning cycle may be required for longer runs and to prevent biofouling. The flux obtained through both PA hollow fibre and CTA flat sheet membranes are less than or equal to 3.15 LMH which is not sufficient for the system to be economical. Improving the performance of FO membranes by reducing the structural parameter through the introduction of nanomodifiers to the membrane material is one of the ways to go. This will enhance the utilization of FO in places where freshwater resources are diminishing and therefore reusing concentrates and reducing their impacts on fresh water sources are of paramount importance.

Round 2

Reviewer 1 Report

After the manuscript was basically revised according to the reviewer's opinion, I think it is acceptable, but some deficiencies are pointed out as follows:

  1. In the page2(L56 and L64), the first line of a paragraph should be indented by two characters.
  2. There are plenty of blank lines on pages 3 to 6.
  3. In the page4(L124 to L64), there are a large number of display errors.
  4. The name of Figure 1 is not standard.
  5. In Figure 2, the position of each small graph is relatively chaotic.